# Ground-Reaction-Force-Based Gait Analysis and Its Application to Gait Disorder Assessment: New Indices for Quantifying Walking Behavior

**DOI:** 10.3390/s22197558

**Published:** 2022-10-06

**Authors:** Ji Su Park, Choong Hyun Kim

**Affiliations:** Center for Bionics, Korea Institute of Science and Technology, Seoul 02792, Korea

**Keywords:** gait analysis, gait disorder, stroke, center of pressure, continuous gait phase, adaptive frequency oscillator, polar gaitogram, area ratio index

## Abstract

Gait assessment is an important tool for determining whether a person has a gait disorder. Existing gait analysis studies have a high error rate due to the heel-contact-event-based technique. Our goals were to overcome the shortcomings of existing gait analysis techniques and to develop more objective indices for assessing gait disorders. This paper proposes a method for assessing gait disorders via the observation of changes in the center of pressure (COP) in the medial–lateral direction, i.e., COPx, during the gait cycle. The data for the COPx were used to design a gait cycle estimation method applicable to patients with gait disorders. A polar gaitogram was drawn using the gait cycle and COPx data. The difference between the areas inside the two closed curves in the polar gaitogram, area ratio index (ARI), and the slope of the tangential line common to the two closed curves were proposed as gait analysis indices. An experimental study was conducted to verify that these two indices can be used to differentiate between stroke patients and healthy adults. The findings indicated the potential of using the proposed polar gaitogram and indices to develop and apply wearable devices to assess gait disorders.

## 1. Introduction

Gait is a pattern of limb movements made during locomotion through space, and maintaining balance is essential for a normal gait. The state in which a normal gait is impossible because of difficulty in maintaining balance is called a gait disorder. Gait disorders are caused by functional impairment or structural lesions in the nervous system, which is involved in gait function. Stroke is a prime example of a disease that causes gait disorder [1,2].

Stroke is a cerebrovascular disease caused by brain damage due to blockage or rupture of blood vessels that supply blood to the brain, which can cause neurological abnormalities, including hemiplegia and impairment of cognitive, motor, and communication functions [3,4]. Stroke-induced motor dysfunction is a disability caused by weakness or hypertonicity of muscles, which increases the risk of hemiplegic gait and falls due to physical imbalance [5,6,7].

Assessment methods for stroke-induced gait disorder often use visual observation of patient movement; however, because such methods rely on the experience of the clinician, it is impossible to quantify the gait disorder. Therefore, studies have focused on the use of wearable devices to develop objective gait disorder assessment methods. Studies have been conducted on gait patterns, with a focus on body acceleration and angular acceleration, using inertial measurement units [8,9,10], observation of myoelectric signals activated during the gait cycle using electromyography [11,12,13,14], and quantitative analysis of gait characteristics, with a focus on the measurement and observation of ground reaction forces (GRF) generated during walking through the attachment of pressure sensors to insoles [15,16,17,18,19,20,21,22].

Among the various studies in which insole devices were used, Hong et al. [16] assessed gait time and speed through a 20 m gait experiment, and Wang et al. [17] used a capacitive pressure sensor to measure indices such as the step count, stride time, phase coordination index (PCI), and percentage of plantar pressure difference (PPD) for gait assessment of stroke patients. Wang et al. [17] confirmed that the step count, PCI, and PPD can be useful indices for assessing the asymmetric gait of stroke patients. Among these indices, stroke patients exhibited a higher step count than healthy adults when they traveled the same distance because of a lower gait speed, and their PCIs and PPDs exhibited instability due to an imbalance between the affected and unaffected feet. Moreover, Seo et al. [18] used insole devices equipped with both a pressure sensor and an inertial sensor to compare the stance time and swing time of the affected and unaffected feet of stroke patients. The results were similar to those of Wang et al. [17], indicating that the asymmetry in the gait phase time for both feet can be used to diagnose stroke patients.

Among them, PCI and PPD are indices for gait analysis using GRF data that have the advantage of enabling wearable devices for real-time gait analysis, because the sensor is inexpensive and the index is easy to calculate. PCI is an index for evaluating gait asymmetry by comparing the stride time between the right and left feet within one gait cycle. The stride time is calculated using the heel contact event [23]. However, when applied to stroke patients, it has the disadvantage of a high error rate due to shuffling [24]. Furthermore, PPD is an index that compares the weight supported by the right and left feet within one gait cycle [25]. However, the reliability of the gait analysis results of most insole devices is low because the pressure sensors are installed only at the main parts of the insole to reduce power consumption [19].

The most significant problem with gait analysis techniques that use insole devices is that changes in the GRF are inconsistent. In the gait analysis technique, as in the studies of Tang et al. [26] and Lim et al. [27], the gait phase is classified by considering the change in GRF or the center of pressure (COP). However, Duong et al. [28] found that anterior–posterior COP in patients with gait disorders showed more significant variability than in healthy adults. It means that the probability of an error may increase if the existing gait phase detection method is used for the patient. It is also explained that this variability in COP is due to the gait strategy of placing the mid-foot on the ground to increase gait stability upon initial contact in patients with gait disorders [29]. According to the study of Jung et al. [30], it was confirmed that the anterior–posterior COP (COPy) appeared more unstable than the medial–lateral COP (COPx) by the gait strategy described above.

When the COP is used for gait phase detection, both the COP in the direction of gait and the direction perpendicular to it, i.e., COPy and COPx, are used [27,31]. However, the COPy of patients with gait disorders exhibits a lack of consistency, as mentioned above; consequently, the success rate of gait phase detection is significantly reduced.

The purpose of the present study was to overcome the problems of existing gait analysis techniques using insole devices and to propose more objective indices for the assessment of gait disorders. 

For the case of a gait phase detection method, it is possible to use the discrete form created by the gait event measurement [32,33]. However, when we apply this detection method to control the walking assistance robot, it can often impart an unwanted ‘load’ rather than an ‘assistant’ to the robot joints, making the robot wearer uncomfortable. Therefore, this study estimated the patient’s gait cycle using the continuous gait phase detection method, such as the adaptive frequency oscillator (AO) with COP data as input data. According to the AO algorithm, the better the consistency of the COP data, the better the accuracy of the gait cycle estimation result. 

To achieve the gait disorder assessment, gait cycles were estimated by using COPx data, which are less significantly affected by gait disorders than COPy data, as input data for the AO algorithm. 

Second, the estimated gait cycles were used to examine the changes in COPx in each gait cycle to develop a new indicator for differentiating between healthy adults and stroke patients, and its performance was tested and compared with that of the aforementioned indices. 

In this study, we introduced a new type of gaitogram, polar gaitogram, representing COP changes in polar coordinates, and a new index, area ratio index (ARI), extracted from and based on this, to propose a more objective gait disorder determination method. 

## 2. Materials and Methods

### 2.1. System Setup

Figure 1 shows the insole device and data collection methods used in this study. The printed circuit board (PCB) shown in Figure 1a was developed for the collection and processing of GRF data measured by pressure sensors and was fabricated to be small and lightweight to minimize its influence on the wearer. The microcontroller unit (MCU) used for processing the GRF data and pressure sensor were STM32F411x (STMicroelectronics, UK) and a force sensing resistor (FSR 402; Interlink Electronics, Inc. CA 93012, USA), respectively. The sensor locations were selected according to our previous study [31]. Details regarding the insole device are presented in Appendix A of [34]. 

As shown in Figure 1b, the GRF measured from the left foot was transmitted to the right-foot MCU to be synchronized with the GRF data from the right foot. The GRF data for both the left and right feet were transmitted to a personal computer, and the collected data were analyzed. The GRF data were collected at a sampling rate of 100 Hz and analyzed using Python.

### 2.2. Subjects and Test Method

The subjects in the gait experiment were all male, including eight healthy adults and four stroke patients (Table 1). None of the subjects in the healthy adult group had neurological disorders, and they were all capable of walking independently. The subjects in the stroke patient group were functional ambulation category level 5 patients who were capable of walking independently without an assistive device. In the stroke patient group, three subjects (subject #9, subject #10, and subject #11) have been left-side hemiplegic for 10, 18, and 4 years after stroke onset of stroke, respectively. Furthermore, the remaining subject (subject #12) has been right-side hemiplegic for 13 years since the stroke onset.

In the gait experiment, the subjects stood facing forward, and upon instruction from the operator to start the experiment, the subjects walked a distance of 5 m on flat ground and stopped at a marked position. Regarding the gait speed, the subjects were instructed to walk at a comfortable pace. The gait experiment was performed 20 times for the healthy adult group and 10 times for the stroke patient group.

The experimental protocol was approved by the Institutional Review Board of the Korea Institute of Science and Technology (KIST). All participants provided written informed consent prior to participation.

### 2.3. Indices for Analysis of Gait Characteristics 

After the changes in COPx according to the gait cycle were drawn with polar coordinates, the areas inside the two closed curves were compared, and the slope of the tangential line at the point where the two closed curves intersected was examined. Thus, a method was developed for differentiating between the gaits of healthy adults and stroke patients. The two proposed indices were tested and compared with the PCI and PPD—existing indices used for stroke diagnosis—to evaluate their potential for the assessment of patients with gait disorders.

#### 2.3.1. Continuous Gait Phase Analysis

In previous studies, gait has been classified into eight discrete phases [31]. However, when a person walks while wearing a gait-assistance robot for gait rehabilitation, it is necessary to operate the robot according to the movement of the wearer by precisely controlling the motor for ensuring a smooth gait [35]. Accordingly, the gait phases must be divided and linked continuously. Not doing so can cause awkward movement by the robot operator or injury from a fall. Therefore, a single gait cycle was set as 2π, and the gait phase was specified within the range of 0–2π. This approach has the advantage of a shorter time delay than discrete gait phase analysis methods.

In previous studies involving continuous gait phase analysis, gait phases were classified within a single stride length, and the computations involved the following four methods: (1) estimating the gait phases of the next step using the gait phases of the previous step according to the initial contact point [36]; (2) defining the angle and angular acceleration of leg joints as a circular orbit relation and deriving the vector direction [37,38]; (3) machine learning [39]; and (4) estimating the frequency of the hip joint angle or GRF using an adaptive frequency oscillator (AO) [40,41,42].

Among these, the first method has the disadvantage of not reflecting the current gait phase information; additionally, it cannot be used for patients with asymmetric gaits. It was not used in this study because errors are caused by shuffling at initial contact, particularly for stroke patients [24]. The second method requires data regarding leg joint movement. In other words, the joint angle must be measured using an encoder to use this method. This method could not be used in the present study, because it employs GRF data. Moreover, machine-learning-based methods are computationally intensive, which makes them difficult to implement in real time for wearable devices. Certainly, there are study cases that inserted a machine learning model into microcontrollers, but this study did not consider this method to minimize the computational burden [43].

Accordingly, in the present study, the gait phase estimation method using an AO employed by Seo et al. was utilized [41]. The concept of this method is shown in Figure 2.

As shown in Figure 2, the AO derives the estimated signal $\hat{u}\left(t\right)$ under the assumption that the input signal u(t) consists of a linear combination of a certain frequency and multiples of this frequency. The following equations are used [40,41,42,44]:(1)$${\dot{\varphi }}_{i}\left(t\right)=i\omega \left(t\right)+k_{\varphi }e\left(t\right)cos\left(\varphi_{i}\left(t\right)\right)\;$$
(2)$$\dot{\omega }\left(t\right)=k_{\omega }e\left(t\right)cos\left(\varphi_{1}\left(t\right)\right)\;$$
(3)$${\dot{\alpha }}_{i}\left(t\right)=k_{\alpha }e\left(t\right)sin\left(\varphi_{i}\left(t\right)\right)\;$$
(4)$${\dot{\alpha }}_{0}=k_{0}e\left(t\right)\;$$
(5)$$\hat{u}\left(t\right)=\alpha_{0}\left(t\right)+\sum^{\;}\alpha_{i}\left(t\right)sin\left(\varphi_{i}\left(t\right)\right)\;$$
(6)$$e\left(t\right)=u\left(t\right)-\hat{u}\left(t\right)\;$$

In Equations (1)–(6), $\varphi_{i}$ and $\alpha_{i}$ represent the ith oscillator phase and amplitude, respectively, $\alpha_{0}$ represents the magnitude of the basic offset of the signal, and ω represents the signal frequency. The AO uses these four variables to compute the trained signal $\hat{u}$. The AO algorithm used in this study was designed as a fifth-order equation (i = 5). Here, $k_{\varphi }$, $k_{\alpha }$, $k_{\omega }$, and $k_{0}$ are the parameters that determine the adaptive speed of the oscillator phase, oscillator amplitude, signal frequency, and oscillator amplitude offset of the input signal, respectively. For healthy adults, $k_{\varphi }=0.8$, $k_{\alpha }=1.2$, $k_{\omega }=0.6$, and $k_{0}=1.0$ were used, and for stroke patients, $k_{\varphi }=0.2$, $k_{\alpha }=0.2$, $k_{\omega }=0.4$, and $k_{0}=0.8$ were used. According to the authors’ research experience, these coefficients are affected by the stiffness and friction coefficient of the insole material, so they should be adjusted according to the experimental conditions.

The GRF data were used to calculate COPx, in accordance with Equation (7). In this equation, $x_{i}$ represents the position of the i^th^ pressure sensor, which is defined by the wearer’s height [34]. Moreover, to assess the COPx value as a ratio relative to the outermost sensor of the foot, it was calculated by dividing the relative position of COPx, i.e., $\sum^{\;}\left(F_{i}\times x_{i}\right)/\sum^{\;}F_{i}$, by the position of the pressure sensor on the fifth metatarsal bone.

Changes in COPx during a single gait cycle (2π) were drawn with orthogonal coordinates and appeared as a sine wave, as shown in Figure 3. Therefore, it was determined that COPx is appropriate for use as an input signal for the AO algorithm; thus, it was input into u(t) of Equation (6) to estimate $\hat{u}\left(t\right)$. The estimated signal $\varphi_{1}$ was derived from the gait phase.
(7)$$COPx=\frac{\sum^{\;}\left(F_{i}\times x_{i}\right)}{\sum^{\;}F_{i}}\times \frac{100}{x_{5th-meta}}\;$$

#### 2.3.2. Area Ratio Index (ARI)

When the orthogonal coordinate graph shown in Figure 3 was converted to polar coordinates using Equations (8) and (9), two closed curves were drawn, as shown in Figure 4. This diagram was denoted as a polar gaitogram.
(8)r=|COPx| 
(9)$$\theta =\varphi_{1}\;$$

According to Equation (7), the upper and lower closed curves in the graph shown in Figure 4 represent the right- and left-foot stance phases with positive and negative values, respectively. With an ideal symmetric gait, the areas of the two closed curves in Figure 4 are identical because the times at which the two feet touch the ground are the same for each gait cycle. However, for actual gait, the areas of the two closed curves differ; thus, the areas were compared for use as a gait analysis index.

The area inside the two closed curves ($Area_{Closed}$) was calculated using Equation (10), and the right- and left-foot stance phases were differentiated with the subscripts R and L to define $Area_{Closed.R}$ and $Area_{Closed.L}$, respectively. Moreover, the ratio of the area of each closed curve to the sum of the areas of the two closed curves and the difference between the ratios were defined as $Area_{ratio.R}$, $Area_{ratio.L}$, and ARI and calculated using Equations (11)–(13), respectively. Here, the ARI represents the difference in weight-bearing time between the two feet. It is equal to 0% if the weight-bearing ratios of the two feet are identical. When the ARI exceeds the threshold value ($ARI_{threshold}$), this indicates that the weight bearing during walking is excessively concentrated on one side. Accordingly, the ARI is proposed as a gait analysis index herein.

If $ARI_{threshold}$ is the threshold value for differentiating between the maximum ARI value of the healthy adult group and the minimum ARI value of the stroke patient group, $Area_{ratio.disorder}$ for differentiating gait disorders can be calculated using Equation (14).
(10)$$Area_{Closed}=\int^{\;}\frac{1}{2}r^{2}\theta d\theta \;$$
(11)$$Area_{ratio.R}=\left(Area_{Closed.R}/\left(Area_{Closed.R}+Area_{Closed.L}\right)\right)\times 100$$
(12)$$Area_{ratio.L}=\left(Area_{Closed.L}/\left(Area_{Closed.R}+Area_{Cloased.R}\right)\right)\times 100$$
(13)$$ARI=\left|Area_{ratio.R}-Area_{ratio.L}\right|\;$$
(14)$$Area_{ratio.disorder}=50\left(\%\right)\pm ARI_{threshold}/2$$

#### 2.3.3. Slope of Tangential Line of Closed Curves 

The straight line drawn by connecting the starting and ending points of the two closed curves becomes the tangential line of the two closed curves and is also a segment that connects the two points where the weight shifts from the left foot to the right foot and from the right foot to the left foot. The slope of the tangential line can be calculated using Equation (15). Here, $r_{R}$ and $\theta_{R}$ represent r and θ, respectively, when COPx changes from positive to negative, and $r_{L}$ and $\theta_{L}$ represent r and θ, respectively, when COPx changes from negative to positive.
(15)$$Ang_{tangent}=\left|atan2\left(\frac{\;\theta_{R}-\theta_{L}\;}{\;r_{R}-r_{L}\;}\right)\right|\;$$

#### 2.3.4. PPD

The PPD was the gait assessment index used in the studies of Wang et al. and Sanghan et al. [17,19]. It reflects the difference in the weights supported by the two feet using the GRF measured during walking.
(16)$$PPD=\frac{2\left|GRF_{R}-GRF_{L}\right|}{GRF_{R}+GRF_{L}}\times 100\;$$

#### 2.3.5. PCI

The PCI was the gait assessment index used in the studies of Wang et al., Rampp et al., and Plontniok et al. [17,45,46]. It reflects the ratio of time for alternate heel strike within a gait cycle, with a single gait cycle set as 2π, i.e., in the sequence of a right heel strike, a left heel strike, and another right heel strike, the ratio of the time required for the second heel strike after the first heel strike to the time from the first heel strike to the third heel strike was examined. The PCI is calculated using the following equations:(17)$$\phi_{i}=2\pi \times \frac{t_{Li}-t_{Ri}}{t_{R\left(i+1\right)}-t_{Ri}}\;$$
(18)$$\phi_{ABS}=\left|\phi_{i}-\pi \right|$$
(19)$$P_{\phi_{ABS}}=100\times \phi_{ABS}/\pi$$
(20)$$\phi^{\prime }=\frac{1}{N}\overset{n}{\underset{i=1}{\sum }}\phi_{i}$$
(21)$$\delta =\sqrt{\frac{1}{N}}\overset{n}{\underset{i=1}{\sum }}{\left(\phi^{\prime }-\phi_{i}\right)}^{2}$$
(22)$$\phi_{CV}=\delta /\phi^{\prime }$$
(23)$$PCI=\phi_{CV}+P_{\phi_{ABS}}\;$$

In Equation (17), $t_{Ri}$ and $t_{Li}$ represent the *i*th heel strike times of the right and left feet, respectively, and the $\phi_{i}$ calculated using these variables is π when the gait is ideally symmetric. Equation (18) was used to calculate $\phi_{ABS}$, i.e., the absolute difference between $\phi_{i}$ and π, and Equation (19) was used to calculate the ratio of $\phi_{ABS}$ to π. To analyze the consistency of gait, the rate of change (coefficient of variation, CV) for the phases of both feet was calculated using Equations (20)–(22). Ultimately, the PCI was determined as the sum of the CV and $P_{\phi_{ABS}}$, in accordance with Equation (23).

## 3. Results

### 3.1. Continuous Gait Phase

Figure 5 shows the continuous gait phase estimation obtained by calculating COPx and applying it to the AO algorithm in the gait experiment on healthy adults. In Figure 5a, the black and red graphs represent COPx used as input data and the $\hat{u}\left(t\right)$ values estimated through training, respectively. The results indicate that the signals estimated by the AO were synchronized with the input signals within a relatively short time from when the subject began walking, i.e., within approximately three steps. Figure 5b shows the obtained changes in the continuous gait phase ($\varphi_{1}$), which represent the frequency components of the estimated $\hat{u}\left(t\right)$. Therefore, it was confirmed that continuous gait phases could be obtained by inputting COPx data into Equations (1)–(6).

Figure 6 shows a scatter plot of the continuous gait phases of a healthy adult (subject #1, Figure 6a) and stroke patient (subject #12, Figure 6b) estimated using the AO algorithm. The results indicate that the gait cycles of healthy adults and stroke patients were continuously estimated within a consistent range. When COPx data were used with the AO algorithm, the gait cycles of patients with gait disorders were successfully estimated.

Table 2 presents the continuous gait phase estimations for all the individuals who participated in the gait experiment. As shown, the gait cycles of the healthy adults and stroke patients were 6.275 ± 0.004 rad and 6.262 ± 0.009 rad, respectively; thus, the results were relatively consistent. Moreover, the gait cycles of healthy adults and stroke patients were shorter than the ideal value of 2π by 0.008 and 0.021 rad, respectively, indicating that patients with gait disorders had slightly shorter gait cycles than healthy adults.

### 3.2. Results for Gait Parameters

Table 3 and Table 4 present four different gait analysis indices obtained by analyzing the experimental data of the healthy adult and stroke patient groups, respectively. 

The ARI—the proposed index for analyzing gait characteristics—represents the ratio of the weight-bearing time of the affected and unaffected feet. ARI values of 2.9% ± 2.6% and 22.1% ± 7.1% were obtained for the healthy adult and stroke patient groups, respectively, indicating that the weight-bearing times were similar between the two feet for healthy adults, whereas for stroke patients, the weight-bearing time of the unaffected foot was approximately 22.1% longer than that of the affected foot. Moreover, the stroke patients exhibited a minimum ARI value of 12.2%, whereas the healthy adults exhibited a maximum ARI value of 6.9%; thus, $ARI_{threshold}$, which is used to differentiate the two groups, can be set as 10% according to the current subject groups. Therefore, according to Equation (14), $Area_{ratio.disorder}$ = 50% + 10%/2, i.e., if $Area_{ratio.R}$ or $Area_{ratio.L}$ is >55%, it can be assumed that a gait disorder has occurred. Moreover, the foot with $Area_{ratio}$ > 55% can be classified as the unaffected foot responsible for a longer weight-bearing time during a single gait cycle, whereas the opposite foot can be classified as the affected foot responsible for <45% of weight bearing. According to this finding and the results presented in Table 4, subjects #9, #10, and #11, for whom $Area_{ratio.L}$ was <45%, were classified as left hemiplegic patients, whereas subject #12, for whom $Area_{ratio.R}$ was <45%, was classified as a right hemiplegic patient. This classification is consistent with the actual statuses of the patients.

Another index proposed herein is $Ang_{tangent}$, which was 0.07 ± 0.05 rad and 0.26 ± 0.11 rad for healthy adults and stroke patients, respectively. The results indicated a difference that could be used to differentiate between the two groups. However, there was only a small difference of 0.03 rad between the healthy adult with the highest $Ang_{tangent}$ value (subject #4; 0.14 rad) and the stroke patient with the lowest $Ang_{tangent}$ value (subject #12; 0.17 rad). When $Ang_{tangent}$ is used to assess gait disorder, the cutoff can be viewed as 0.15–0.16 rad, and a value exceeding the threshold can be classified as a gait disorder.

Accordingly, the ARI and $Ang_{tangent}$ values can be used to differentiate between healthy adults and stroke patients. 

### 3.3. Comparison with Other Parameters

Table 3 and Table 4 present the results for the PPD and PCI, i.e., indices used in previous studies for analyzing gait characteristics, which were obtained using GRF data in the present study. 

The PPDs for healthy adults and stroke patients were 7.77% ± 1.51% and 22.71% ± 10.15%, respectively, indicating a clear difference in the mean values. A PPD of 10% can be set as the threshold, and values exceeding this threshold can be used to identify patients with gait disorders who have more weight concentrated on one side.

The PCIs for healthy adults and stroke patients were 9.26% ± 2.68% and 13.81% ± 0.59%, respectively. However, two healthy adults (subjects #2 and #6) exhibited values close to 12–13% (a level similar to that of the stroke patients); thus, it was determined that gait disorders cannot be differentiated using the PCI.

Figure 7 shows comparisons of the ARI, PPD, and PCI, which are gait analysis indices calculated for healthy adult and stroke patient groups. 

As shown in Figure 7, the ARI has sufficient discriminatory power to be used as a gait analysis index. With $ARI_{threshold}$ = 10%, the healthy adult and stroke patient groups were clearly divided relative to the threshold, indicating that the proposed ARI can be used as a gait analysis index. The results also indicated that the PPD with a threshold of 10% can be used as a gait analysis index.

We performed the Mann–Whitney U test to compare the ARI, PPD, and PCI levels of healthy adults and stroke patients and set all statistical significance levels to 0.05. According to the analysis results, the *p*-values of ARI, PPD, and PCI were 0.004, 0.008, and 0.008, respectively, which showed significant differences, confirming that all of these indices can distinguish between healthy adults and the stroke group.

However, for the PCI, healthy adults and stroke patients overlapped near the threshold, indicating that it cannot be used as a gait analysis index.

Figure 8 and Figure 9 show the polar gaitograms obtained in the present study for the healthy adult and stroke patient groups, respectively. The COPx indicating the right-foot stance phase as a positive number (upper closed curve) and the left-foot stance phase as a negative number (lower closed curve) are shown in blue and red, respectively.

In the polar gaitogram of healthy adults shown in Figure 8, the areas inside the upper and lower closed curves appear to be similar. In the polar gaitogram of stroke patients shown in Figure 9, the areas inside the upper and lower closed curves differ significantly. These results confirm that the polar gaitogram can be used to demonstrate that the two groups have significantly different $Area_{ratio}$ values, as shown in Table 3 and Table 4, and the side with a smaller area inside the closed curve can be visually identified as the affected side.

Moreover, as shown in Figure 9, the angle formed by the line segment shown as a dashed line and the horizontal line at the center of the polar gaitogram ($Ang_{tangent}$) was larger for the stroke patients than for the healthy adults.

## 4. Discussion

The PPD and PCI are parameters used in previous studies for monitoring the statuses of stroke patients using an insole device or force plate. These parameters were used to compare the magnitudes and durations of weight bearing in the affected and unaffected feet.

In a study performed by Wang et al. [17], both the PPD and PCI were used in gait assessment for stroke patients, whereas Sanghan et al. [19] used only the PPD. Additionally, the insole devices used in these studies had different numbers and locations of sensors for measuring the GRF. Wang et al. [17] used 10 pressure sensors (2 on the toes, 4 at the metatarsal position, and 4 at the heel) on each foot, as shown in Figure 10b, whereas Sanghan et al. [19] used Pedar-x (Novel GrmbH, Munich, Germany) with a total of 99 pressure sensors, as shown in Figure 10c, for measuring the GRF.

In the present study, the minimum number of pressure sensors—five sensors (one each on the big toe, first metatarsal bone, fifth metatarsal bone, cuboid bone, and heel)—on each foot was used, as shown in Figure 10a.

To assess the effect of the number of pressure sensors used in the insole device on the analysis of the gait characteristics, the mean PPD and PCI values from the three studies were compared, as shown in Table 5. Although the PPD and PCI values were obtained using different insole devices, the results of all three studies indicated that the PPD and PCI had sufficient discriminatory power to differentiate between healthy adults and patients with gait disorders. 

There were differences in the values, and to identify the cause of such differences, Pedar-x, which was used by Sanghan et al. [19], was used in the present study to measure the PPD and PCI again for only healthy adults, and the results were compared, as shown in Table 5. The KIST (w/Pedar-x) data exhibited PPD values that were similar to those reported by Wang et al. [17] and Sanghan et al. [19]. However, the PPD measured by the insole device developed in the present study was 44%–55% larger than those of the other studies, despite the use of fewer sensors. Moreover, the PCI measured in the present study was 65% and 275% larger than those of Wang et al. and KIST (Pedar-x), respectively.

This difference appears to be due to the different numbers and positions of pressure sensors. In general, as the number of pressure sensors attached to the insole increases, the total load measured by the pressure sensors tends to be closer to the actual body weight of the subject. In this case, according to Equation (16), the ratio of the total body weight to the weights measured in the left and right feet decreased as the number of pressure sensors increased. Therefore, because the number of pressure sensors used in the present study was far smaller than those in other studies, the PPD values were always larger.

Meanwhile, a comparison of the positions of the pressure sensors used in the present study and other studies indicated that the pressure sensors were concentrated at the center of the bottom of the foot in the present study. Considering the width of the foot (not the length), all the sensors were located inside the foot. Examining the GRF signals in such cases revealed that the sensor located on the heel was pressed after the foot struck the ground fully; thus, it was pressed later than in other studies. Meanwhile, the sensor on the big toe was not pressed during walking for some subjects. Consequently, the CV value that represents the gait consistency, which is given by Equation (22), increased, which is suspected to have contributed to the higher PCI value compared with other studies.

Generally, increasing the number of pressure sensors used in an insole device increased the computational accuracy of the gait analysis index derived using GRF data. However, increasing the number of sensors also increases the computational burden, which leads to increased power consumption and significantly reduces the length of continuous use of the insole device. Therefore, the number of pressure sensors should be minimized within the range that allows the gait characteristics to be analyzed.

In this study, FirmTech’s Bluetooth chip (FB755AS, Firmtech Co., Seongnam-si, Korea) was used to collect and transmit GRF data. According to the manufacturer’s datasheet, there is a wireless delay time of up to 10 ms. Therefore, we investigated the time delay in data acquisition. The size of the collected gait experiment data based on one stride for a healthy adult (subject #1) and a stroke patient (subject #9) are 64.1 ± 2.7 data packets (641 ± 27 ms) and 175.9 ± 12.4 data packets (1759 ± 124 ms), respectively. Therefore, in the case of subject #1 and subject #9, the time delay that may occur in the walking experiment is 1.6% and 0.6%, respectively. It is an outstanding result, as there is no problem in real-time data acquisition and analysis. 

The insole device used in the present study had fewer pressure sensors for measuring the GRF than those used in previous studies. Despite this, the mean values of the PPD- and PCI-derived gait analysis indices could be used to differentiate between healthy adults and stroke patients. However, the PCI values of some healthy adults and stroke patients overlapped near the threshold. Therefore, the application of the PCI to the insole device developed by the authors can reduce the success rate of gait disorder diagnosis.

The proposed polar gaitogram indicated the time for which each foot was responsible for weight bearing as two close curves (upper and lower curves), allowing visual confirmation of the weight-bearing responsibility of each foot, as well as easy identification of the affected foot. Moreover, the ARI value, which was calculated as the difference between the areas inside the two closed curves of the polar gaitogram and exceeded the threshold, could be used as a criterion for the diagnosis of gait disorder. Furthermore, as indicated by Table 3 and Table 4 and Figure 7, the gait disorder diagnostic accuracy achieved using the ARI was higher than that achieved using the PPD and PCI.

In summary, if the ARI value calculated by Equation (13) exceeds the threshold value, the subject can be determined as a patient with gait disorder whose weight is excessively shifted to one foot. In the case of a patient with gait disorder, the foot whose $Area_{ratio.disorder}$ value calculated by Equation (14) exceeds 55% is considered as an unaffected foot.

In addition, $Ang_{tangent}$—the slope of the tangential line of the two closed curves—was found to be an index that can be used to differentiate between healthy adults and stroke patients. However, the $Ang_{tangent}$ values of some subjects in the two groups were near the threshold; thus, it is necessary to establish more specific criteria through additional experimental studies.

## 5. Conclusions

Previous studies that evaluated the presence or absence of gait disorder using an insole device found that the higher the number of pressure sensors, the higher the accuracy of distinguishing between the two groups of healthy adults and patients with a gait disorder. However, they have the disadvantage that power consumption and computation burden also increase with the number of pressure sensors.

In the present study, an insole device with fewer pressure sensors than those used in existing studies was employed to differentiate patients with gait disorders, and the following results were obtained. 

First, COPx data, which were less significantly influenced by gait disorder than COPy data, were input into the AO algorithm to successfully execute continuous estimation within the gait cycle range of (0, 2π). Moreover, a gait experiment was conducted to confirm that this gait cycle estimation method can be applied to both healthy adults and stroke patients.

Second, new gait analysis indices were proposed for differentiating patients with gait disorders. Graphs were drawn with polar coordinates using the aforementioned gait cycle and COPx data, and a method for examining polar gaitograms was developed. Additionally, it was confirmed that the ARI, i.e., the difference between the areas inside the two closed curves in the polar gaitogram, and $Ang_{tangent}$, i.e., the slope of the tangential line in contact with both closed curves, could be used to differentiate between healthy adults and stroke patients. 

When the insole device proposed in this study and the new index ARI were applied, the average ARI value of the group of patients with gait disorders was about 7.6 times that of the healthy adult group. It facilitated distinguishing them from each other, and the ARI values of the two groups did not overlap. This result shows that the ARI performs better in evaluating the presence of gait disorder than PCI or PPD applied with conventional insole devices.

The insole device used in the present study had 50% fewer pressure sensors than those used in previous studies. Consequently, it consumed significantly less power, allowing operation for a longer duration, which can be useful in the development and application of wearable devices for gait analysis. In future studies, the proposed gait analysis indices can be used for assessing the gait characteristics of patients with Parkinson’s disease and knee arthritis and can be used to develop technology for assessing the improvement of gait characteristics after gait rehabilitation for patients with gait disorders.

## Figures and Tables

**Figure 1 sensors-22-07558-f001:**
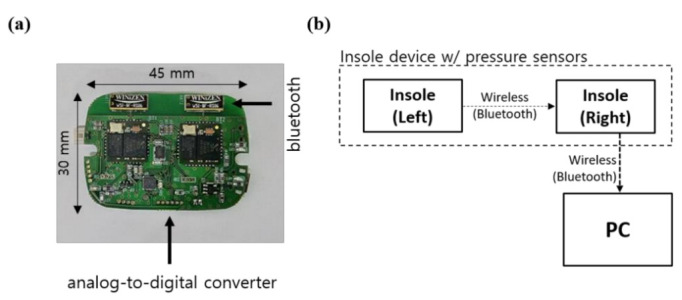
Structure of the smart insole’s PCB. (**a**) Photograph and dimensions of the PCB. (**b**) Schematic of the data measurement system.

**Figure 2 sensors-22-07558-f002:**
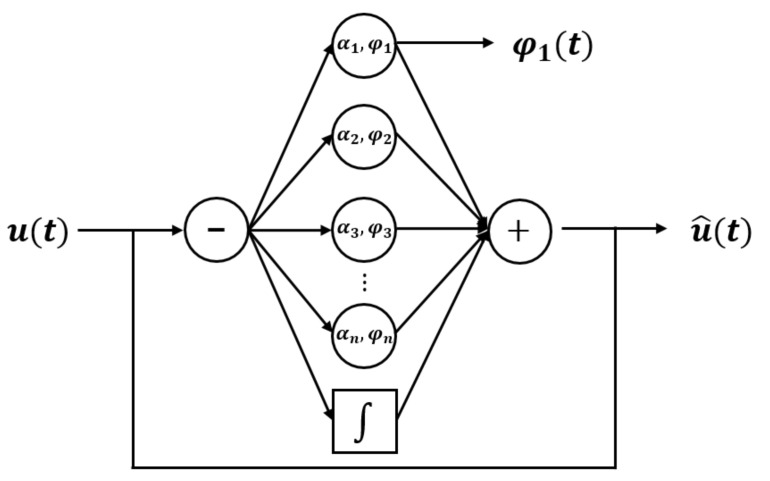
Block diagram of the typical AO.

**Figure 3 sensors-22-07558-f003:**
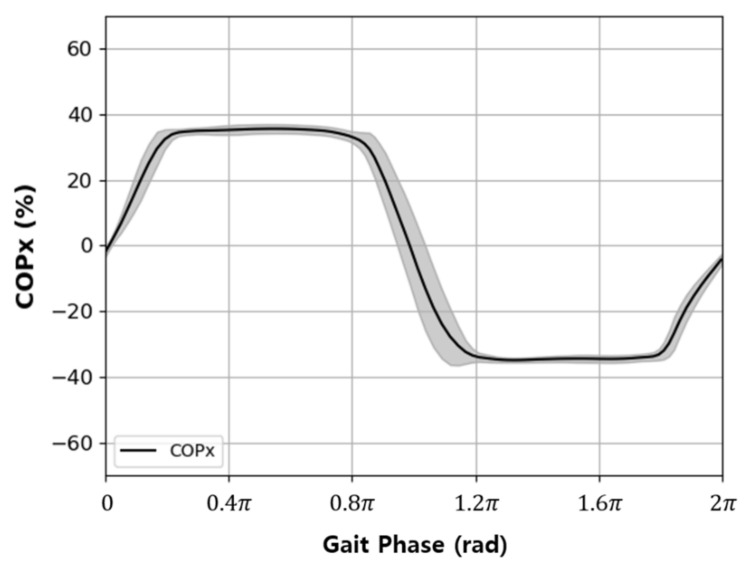
Typical variation in the medial–lateral direction COP data (COPx) with respect to the gait phase in a level walking test.

**Figure 4 sensors-22-07558-f004:**
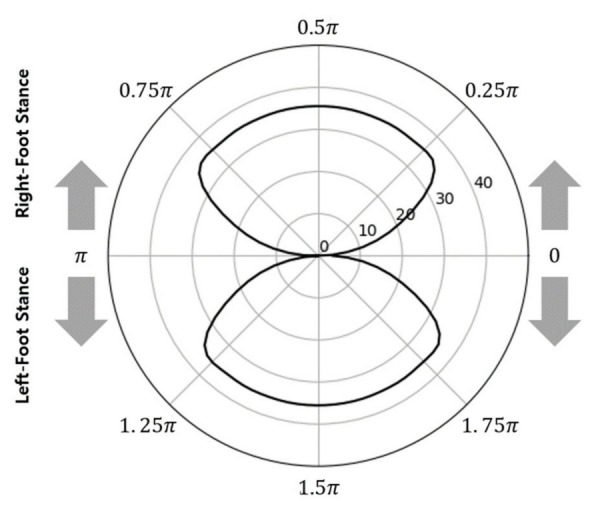
Typical polar gaitogram: gait analysis data expressed in the polar coordinate system with COPx and the gait phase.

**Figure 5 sensors-22-07558-f005:**
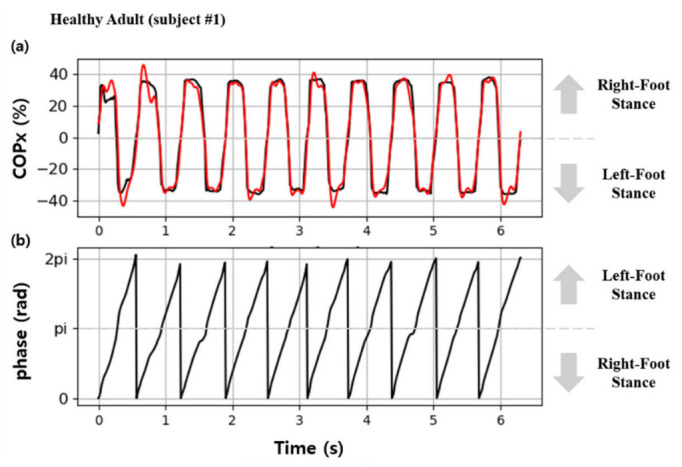
Results of gait phase estimation using the AO: (**a**) the red line indicates the estimated COP, and the black line indicates the input data for the AO; (**b**) estimated continuous gait phase.

**Figure 6 sensors-22-07558-f006:**
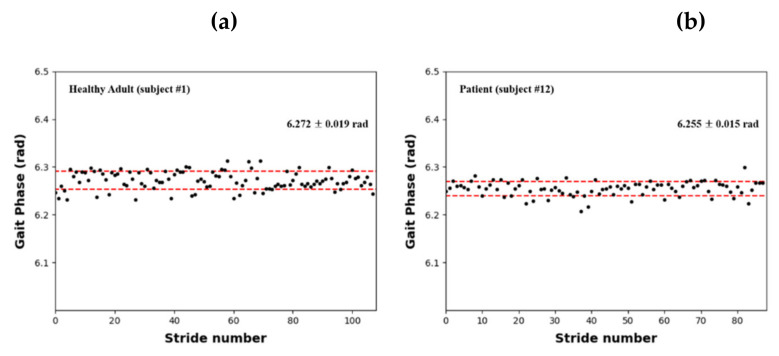
Estimated continuous gait phase results based on the AO with COPx: (**a**) healthy adult (subject #1); (**b**) stroke patient (subject #12).

**Figure 7 sensors-22-07558-f007:**
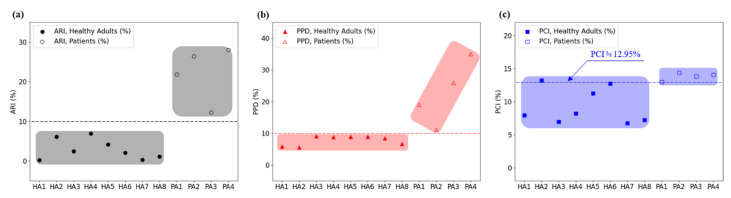
Comparison of the gait disorder classification results obtained using three different indices: (**a**) the ARI, (**b**) PPD, and (**c**) PCI.

**Figure 8 sensors-22-07558-f008:**
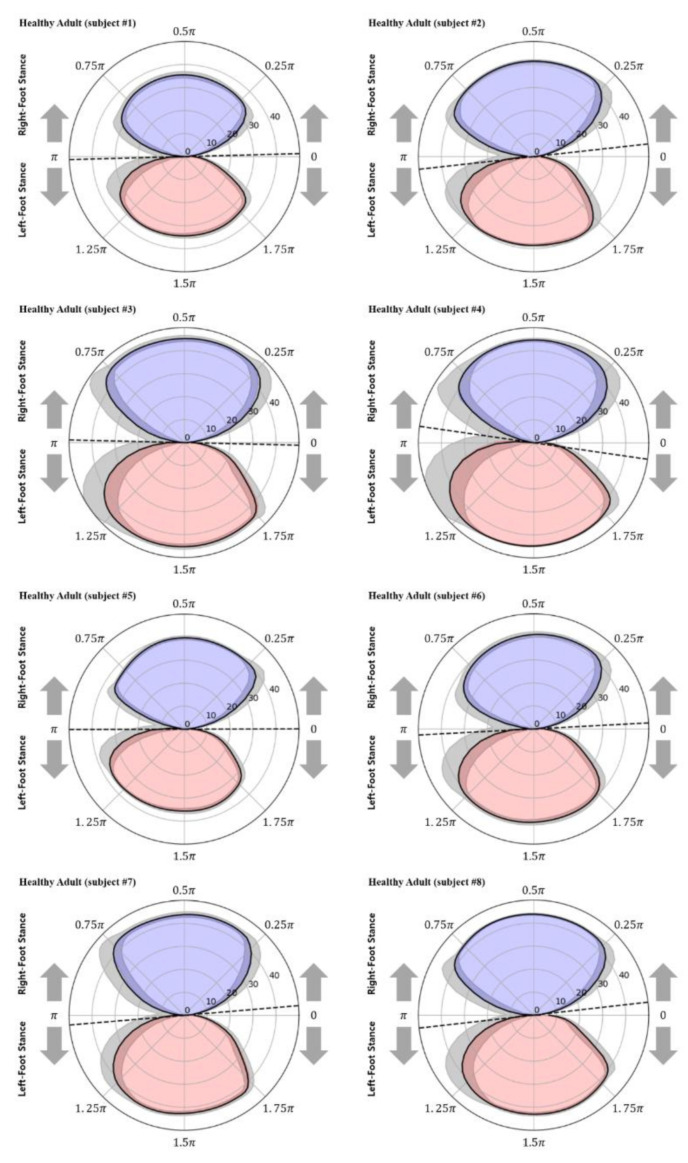
Polar gaitograms for the healthy adult group: blue area, right-foot stance; red area, left-foot stance; gray shadow, standard deviation of COPx; black dotted line, tangential line to both curves.

**Figure 9 sensors-22-07558-f009:**
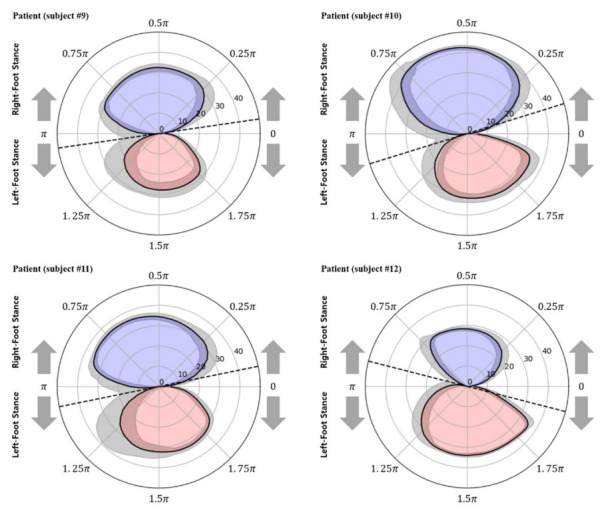
Polar gaitograms for the stroke patient group: blue area, right-foot stance; red area, left-foot stance; gray shadow, standard deviation of COPx; black dotted line, tangential line to both curves.

**Figure 10 sensors-22-07558-f010:**
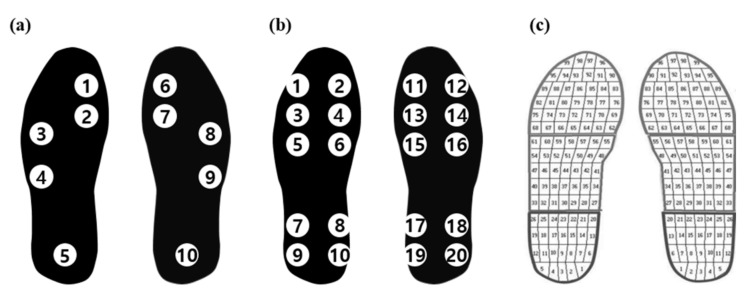
Comparison of the positions and numbers of pressure sensors used in different studies: (**a**) Park et al. 2021; (**b**) Wang et al. 2019; (**c**) Sanghan et al. 2015.

**Table 1 sensors-22-07558-t001:** Subject information.

	Healthy Adults	Patients
Number of persons	8	4
Age (years)	33 ± 3	56 ± 10
Height (m)	1.76 ± 0.05	1.67 ± 0.05
Weight (kg)	70 ± 12	68 ± 6

**Table 2 sensors-22-07558-t002:** Results of the continuous gait phase estimation.

Healthy Adult	Gait Phase (Rad)	Patient	Gait Phase (Rad)
Subject 1	6.272 ± 0.019	Subject 9	6.273 ± 0.017
Subject 2	6.265 ± 0.016	Subject 10	6.255 ± 0.016
Subject 3	6.279 ± 0.025	Subject 11	6.263 ± 0.016
Subject 4	6.275 ± 0.022	Subject 12	6.255 ± 0.015
Subject 5	6.276 ± 0.016	-	-
Subject 6	6.275 ± 0.016	-	-
Subject 7	6.277 ± 0.018	-	-
Subject 8	6.278 ± 0.019	-	-
Average (S.D.)	6.275 ± 0.004		6.262 ± 0.009
Difference with 2π	0.008		0.021

**Table 3 sensors-22-07558-t003:** Gait disorder classification indices for healthy adults.

Healthy Adult Subject No.	$Area_{ratio.R}$ (%)	$Area_{ratio.L}$ (%)	ARI (%)	$Ang_{tangent}$ (rad)	PPD (%)	PCI (%)
Subject 1	49.9	50.1	0.2	0.03	5.86	7.93
Subject 2	53.1	46.9	6.1	0.11	5.50	13.18
Subject 3	48.8	51.2	2.5	0.02	9.09	6.95
Subject 4	46.5	53.5	6.9	0.14	8.84	8.18
Subject 5	52.1	47.9	4.2	0.00	8.89	11.23
Subject 6	48.9	51.1	2.1	0.05	8.89	12.71
Subject 7	50.2	49.8	0.3	0.08	8.44	6.71
Subject 8	50.5	49.5	1.1	0.11	6.65	7.20
Average (S.D.)	50.0 ± 2.0	50.0 ± 2.0	2.9 ± 2.6	0.07 ± 0.05	7.77 ± 1.51	9.26 ± 2.68

**Table 4 sensors-22-07558-t004:** Gait disorder classification indices for stroke patients.

Stroke Patient Subject No.	$Area_{ratio.R}$ (%)	$Area_{ratio.L}$ (%)	ARI (%)	$Ang_{tangent}$ (rad)	PPD (%)	PCI (%)
Subject 9	60.9	39.1	21.8	0.21	18.97	12.97
Subject 10	63.2	36.8	26.4	0.42	11.05	14.35
Subject 11	56.1	43.9	12.2	0.24	25.89	13.85
Subject 12	36.0	64.0	28.0	0.17	34.92	14.05
Average (S.D.)	54.0 ± 12.4	46.0 ± 12.4	22.1 ± 7.1	0.26 ± 0.11	22.71 ± 10.15	13.81 ± 0.59

**Table 5 sensors-22-07558-t005:** PPD and PCI results obtained with different insole devices.

	Healthy Adults		Stroke Patients	
	PPD (%)	PCI (%)	PPD (%)	PCI (%)
KIST, w/smart insole	7.77	9.26	22.71	13.81
KIST, w/Pedar-x	5.41	2.47	-	-
Wang et al. [17]	5.01	5.62	15.90	19.50
Sanghan et al. [19], w/Pedar-x	5	-	30	-

## Data Availability

The data presented in this study are available upon request from the corresponding author. The data are not publicly available, owing to ethical concerns, as they were obtained in a clinical trial.

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
