# Peer review of "Ground-Reaction-Force-Based Gait Analysis and Its Application to Gait Disorder Assessment: New Indices for Quantifying Walking Behavior"

_sensors, 2022, doi:10.3390/s22197558_

Round 1
Reviewer 1 Report
Thank you for submitting your important work. I have few suggestions.
Introduction: 1) From line 25-30, please cite it and I would suggest rephrasing it.
2) In line 49-50, abbreviation for PPD has not been provided. Also, do you want elaborate on PCI and PPD over here, in terms of definition and citations.
Also, why did you include PCI and PPD like terms, is their significance of including these over here? If GRF is focus in this paper and COP analysis, then shouldn't the focus be these terms. Please look into these.
Methods: I have concerns about sample size and age. The young healthy adults are compared with older patients with stroke. If you have more data collected meanwhile, I would suggest including that. It will make this paper stronger, also try to do age and gender match for your population.
I see that you are using PCI and PPD for analysis in methodology, please provide reasoning and explanation of that either in introduction or methodology or both. I would suggest include sub section of your outcome measures.
You did a good job on methodology and results overall.
Discussion: 1) In line 376-379, please cite again the papers from the table. I would suggest wherever you are using Wang and Sanghan et al, please cite it.
2) I think this line has been repeated about the fewer pressure sensors used in the study than the previous studies. Please review it and put it in one paragraph only, either in limitations or discussion. Also, cite other studies that you are mentioning everywhere, citations are missing in this paper.
3) Review line 419-421.
In the end, this paper has potential but need lot of organization and revision.
Reviewer 2 Report
Dear Authors,
The presented article deals with new Indices for Quantifying Walking Behavior Overall usefull in Gait-Disorder Assessment. The article is fairly well-written and focuses on the given issue. The authors work with a sufficient number of professional sources that reflect the chosen issue of the research investigation.
The study is well conducted and the results are relevant in this field. Therefore, I recommend the acceptance for publication.
However, I have only one recommendation, which potentially might contribute to improve the quality of the manuscript.
In the conclusion there is more need to explain the relevance of the results for the clinical practice also emphasizing the novelty compared to other existing indices.
Reviewer 3 Report
In this paper, the authors used an insole device with five force sensors to analyze the walking gait cycle, as as to differentiate walking patterns between healthy subjects and stroke patients. Multiple evaluation indices were applied and compared for the differentiation purpose. Eight healthy subjects and four stroke patients were recruited for overground walking experiments to analyze these evaluation indices. Overall, the method and results in the current manuscript are interesting. However, the writing of the manuscript is poor and needs to be significantly improved. The motivations and innovations of the paper were not well explained. What are the challenges of the current state-of-art to analyze gait patterns by using GRF? What are the benefits of using the medial-lateral COP compared to other COP studies? More specific comments from the reviewer can be found in the attached document.

Round 2
Reviewer 1 Report
Thank you for editing the manuscript based on the comments provided. It made the manuscript stronger.
Author Response
Your comments and suggestions have helped improve the quality of the manuscript. Thank you very much.
Reviewer 3 Report
The reviewer thanks the authors for taking the time to address these comments raised in the first round. Although many issues have been handled successfully, there are still some points that were not addressed well and require more improvement. More specific comments from the reviewer can be found in the attached file.

Author Response
: Your comments and suggestions have helped improve the quality of the manuscript. Thank you very much.
: Please refer to the attached document for our answers.
